# One-Dimensional Maximum Power Point Tracking Design of Switched-Capacitor Charge Pumps for Thermoelectric Energy Harvesting

**Koichi Nono and Toru Tanzawa ***

Graduate School of Integrated Science and Technology, Shizuoka University, Hamamatsu 432-8561, Japan
* Correspondence: toru.tanzawa@shizuoka.ac.jp

**Abstract:** This paper proposes a one-dimensional (1D) maximum power point tracking (MPPT) design which only requires measurement of one parameter (the input voltage of a switched-capacitor charge pump) for calibrating a power converter including the charge pump and thermoelectric generator. The frequency of the clock to drive the charge pump is designed to minimize the circuit area of the entire charge pump circuit for generating a target output current at a specific output voltage. The ratio of the capacitance value of each boosting capacitor (C) to the size of the switching MOSFET can be determined to maximize the transferring current at the same time. When a thermoelectric generator (TEG) is given, its output impedance is determined. Its open-circuit voltage varies with the temperature difference between two plates of the TEG. MPPT maximizes the output power of the charge pump even when the temperature difference varies. It was indicated that the number of stages of charge pump (N) needs to increase when the temperature difference lowers, whereas C needs to decrease inversely proportional to N, meaning that the C–N product should be kept unchanged for MPPT. Demonstration of the circuit design was conducted in 65 nm CMOS, and the measured results validated the concept of the 1D MPPT.

**Keywords:** one-dimensional; maximum power point tracking; charge pump; thermoelectric generator; energy harvesting

## 1. Introduction

More and more Internet of things (IoT) devices are being connected to each other around the globe for a safer society and highly efficient healthcare, agriculture, and industries [1,2]. IoT devices to be placed somewhere with no alternating current (AC) main need to have batteries for powering. Rapid increases in the cost for replacing wasted batteries and in the amount of waste are becoming problematic. Energy-harvesting technology is expected to solve such an economic and environmental challenge by powering IoT devices with environmental energy sources such as lights, vibration, and heat flow [3,4]. Thermoelectric generators (TEGs) generate electric power with heat flow or a temperature gradient [5–7]. Powering sensors with heat flow from heat pipes to air to monitor surrounding temperature and other physical properties is used in chemical plants and fabs [8]. Wearable electronic devices can also work with TEG from body temperature without batteries [9].

Because the nominal open-circuit voltage of TEG ($V_{OC}$) is below 1 V, boost converters are needed to drive sensor integrated circuits (ICs). Switched capacitor charge pumps (CPs) [10] are used, especially in applications which require a small form factor and low power. The design challenge is how high power conversion efficiency can be maintained, namely, maximum power point tracking (MPPT), over wide variations in temperature difference between the two plates of TEG ($\Delta T$) or, in other words, over wide variations in $V_{OC}$ because $V_{OC}$ is proportional to $\Delta T$. Figure 1 illustrates a general power supply system composed of TEG and CP for sensor ICs, as shown in [11,12]. The design parameters of CP

are the stage capacitance $C$, the number of stages $N$, the size of charge transfer switches $W$, and the clock frequency $f$. Those parameters are determined by a given condition for the input voltage $V_S$, output voltage $V_{PP}$, and current $I_{PP}$. When $V_S$ varies according to $V_{OC}$, one or more design parameters need to be varied for the CP to operate at or around the maximum power point. In previous designs [13–15], multidimensional MPPT was proposed and evaluated. However, a greater circuit area was needed to have largely flexible input impedance of CP.

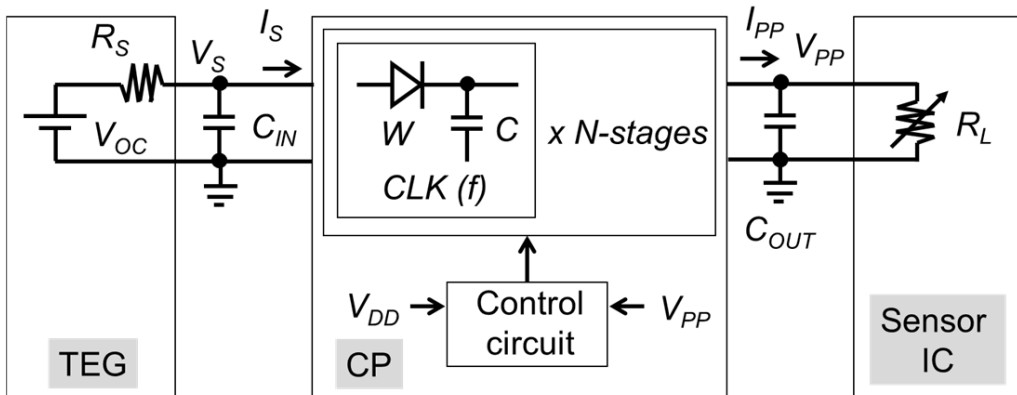

**Figure 1.** Circuit diagram of CP with TEG as a power source and sensor IC as a load.

This paper is aimed at proposing and validating a one-dimensional MPPT to minimize the area overhead of CP even with MPPT capability by applying reconfigurability for CP. This paper is organized as follows: Section 2 reviews previous studies on CP with MPPT and reconfigurable CP in detail. The concept of a reconfigurable charge pump toward maximum output power density is proposed in Section 3. The circuit design is demonstrated in Section 4. Section 5 compares the proposed 1D MPPT design with the previous 2D or 3D ones.

## 2. Previous Work on TEG-CP System with MPPT and Reconfigurable CP

### 2.1. 2D (N, f) MPPT [13]

Figure 2 illustrates a 2D MPPT algorism [13]. The steps to determine the optimum N are as follows:

1. (Step 1) $N$ is set to be the maximum assuming $V_{OC}$ is at the minimum (otherwise, the output voltage cannot reach the target output voltage $V_{PP\_TGT}$). $f$ is set to be the minimum for having room to increase the input power to the CP with faster $f$ during the following searching procedure.
2. (Step 2) CP runs in a predetermined period $Tp$. The peak output voltage is measured as $V_{PP\_past}$.
3. (Step 3) CP runs in $Tp$ with a decreased $N$. The peak output voltage is measured as $V_{PP\_now}$.
4. (Step 4) $V_{PP\_now}$ is compared with $V_{PP\_past}$. If $V_{PP\_now} > V_{PP\_past}$, then Step 3 is done. Otherwise, the procedure moves on to Step 5.
5. (Step 5) $N$ is considered optimum at the current $V_{OC}$, which makes the CP to output the maximum $I_{PP}$.

Then, the steps to determine the optimum $f$ are as follows:

6. (Step 6) CP runs in $Tp$ with an increased $f$. The peak output voltage is measured as $V_{PP\_now}$.
7. (Step 7) $V_{PP\_now}$ is compared with $V_{PP\_past}$. If $V_{PP\_now} > V_{PP\_past}$, then Step 6 is done. Otherwise, the procedure stops.

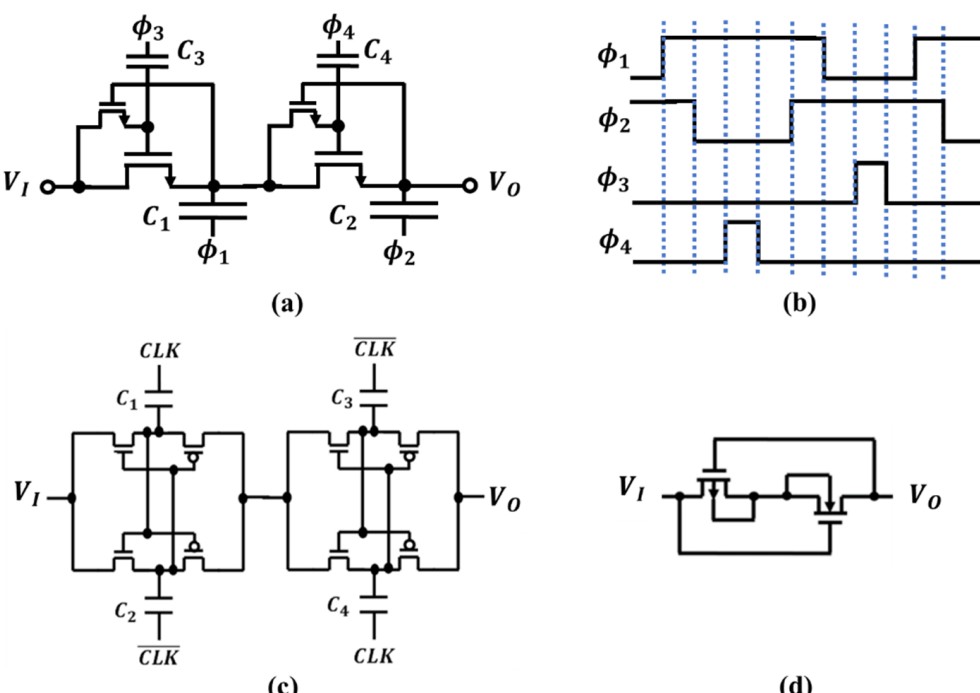

**Figure 2.** Charge transfer switches (CTSs) with local gate boosters (**a**), four-phase clock (**b**), CMOS latch or cross-coupled CSTs (**c**), and ULPD (**d**).

The value of $N$ right after Step 5 can be optimum as long as $V_S$ stays at the value at Step 5. However, $V_S$ decreases as $f$ increases because an increased input current decreases $V_S$ from an IR drop in the output impedance of TEG ($R_S$). Therefore, the optimum $N$ needs to depend on $f$. In order for the procedure to run the CP at the MPP regardless of $V_{OC}$, a fully 2D MPPT would be needed to scan ($N, f$) points on the $N$–$f$ plane. When the numbers of possible $N$ and $f$ are $N_N$ and $N_f$, respectively, one needs to run the CP with different combinations of $N_N \times N_f$ in the worst case, which can take significant time to determine the MPP.

### 2.2. 2D (N, f) MPPT Algorism for CP System with a Supercapacitor and a Linear Regulator [14]

In [14], another 2D MPPT was proposed for the CP system with a supercapacitor and a linear regulator. MPPT is performed only during ramping up the output voltage of CP ($V_{PP}$). While $V_{PP}$ is ramping up, the clock frequency $f$ and the number of stages $N$ are controlled independently. $f$ is controlled in such a way that the input voltage of CP ($V_S$) is around a target voltage ($V_{MPPT}$) for MPPT. For example, when the energy transducer is TEG, $V_{MPPT}$ is $V_S/2$. As $f$ increases, the input impedance of CP decreases; therefore, $V_S$ decreases, or vice versa. $N$ is controlled in such a way that $V_{PP}$ reaches a target voltage $V_{PP\_TGT}$. CP can generate $V_{PP\_TGT}$ even with a low $V_S$ when $N$ is sufficiently large. In other words, at the beginning of ramping up, $N$ is controlled to be sufficiently large. As $V_{PP}$ is approaching $V_{PP\_TGT}$, $N$ is decreased to the number of stages, which is barely sufficient under a given condition of $V_{OC}$. Thus, MPPT is realized at the interface between TEG and CP rather than that between CP and the load.

### 2.3. 3D (C, N, f) MPPT Algorism [15]

In [15], a 3D MPPT was proposed with three design parameters $f$, $C$, and $N$ controlled. In the first MPPT step, $f$ is set at the maximum of 4.25 MHz. $N$ is initially set to be the minimum value. Under a certain load condition, CP is run. Because the input impedance is minimum with the smallest $N$ and the largest $C$, the input voltage of CP ($V_S$) is expected to be lower than the target voltage of $V_{OC}/2$ in case of TEG. $V_S$ is monitored to see if $V_S > V_{OC}/2$. Until $V_S > V_{OC}/2$, $N$ is increased and $C$ is decreased. Note that the input

impedance can vary when a load varies over time; therefore, the CP configuration in terms of $N$ and $C$ may not depend only on $V_{OC}$. Once $N$ and $C$ are determined as an MPPT configuration, $f$ is controlled in such a way that $V_{PP}$ stays at a target voltage $V_{PP\_TGT}$ under the given load condition. As a result, even with 3D MPPT, the final combination of $C$, $N$, and $f$ may not achieve MPPT at the CP output.

*2.4. Reconfigurable CP*

Various charge transfer switches (CTSs) have been proposed to reduce the effective threshold voltage ($V_{TH}$) per CTS. Umezawa et al. proposed effectively zero $V_{TH}$ CTS by using a four-phase clock [16], as shown in Figure 2a,b. $C_1$ and $C_2$ are the stage capacitors which mainly determine the circuit area. The other capacitors can be small, which aim at boosting the gate of CTSs. After the gate node is left floating, the small gate-boosting capacitors $C_3$, $C_4$ allow the transfer transistors to operate in triode region, resulting in effectively zero $V_{TH}$. Gariboldi et al. proposed CTS with a CMOS latch or cross-coupled CMOS with two-phase clock [17], as shown in Figure 2c. A stage capacitor is halved for each of the two capacitors $C_1$, $C_2$ to remain the same stage capacitance in total. Charges can be fully transferred from $C_1$ to $C_3$ with CLK high and /CLK low in the first half period. Charges can be fully transferred from $C_2$ to $C_4$ with CLK low and /CLK high in the second half period. As a result, the same amount of charge can be transferred from one stage to the next in one clock cycle. Levacq et al. proposed an ultralow-power diode (ULPD) [18], as shown in Figure 2d. In a forward biasing condition, either the NMOSFET or the PMOSFET with a lower threshold voltage determines the forward bias current. In a reverse biasing condition, a significant reduction in off-leak current is expected.

To improve the power conversion efficiency of the RF-DC converter for RF energy harvesting over a wide input power range, a reconfigurable CTS was proposed in [19], as shown in Figure 3. Selectors can connect the gates of PMOSFETs with those of NMOS to be configured as a CMOS latch, as shown in Figure 3a, in a relatively low-input-power condition where the forward bias current is prioritized rather than low reverse leakage. CTS can be reconfigurable as a hybrid topology, as shown in Figure 3b, in a relatively high input power condition where low reverse leakage is prioritized rather than the forward bias current.

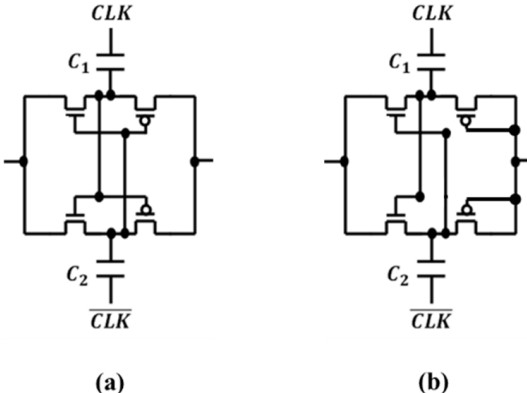

(a)  (b)

**Figure 3.** Reconfigurable CP with variable topology of CTSs; CMOS latch CTS in a low input power range (**a**) and a hybrid CTS n a high input power range (**b**).

Figure 4a shows the schematic diagram of two-stage unit based on the structure of Figure 2a. Control signals *ENP1*, *ENP2*, and *ENS* determine the charge transfer path among IN-to-OUT_S, IN-to-OUT_P, IN_S-to-OUT_S, and IN_S-to-OUT_P. Figure 4b illustrates a symbol of the two-stage unit.



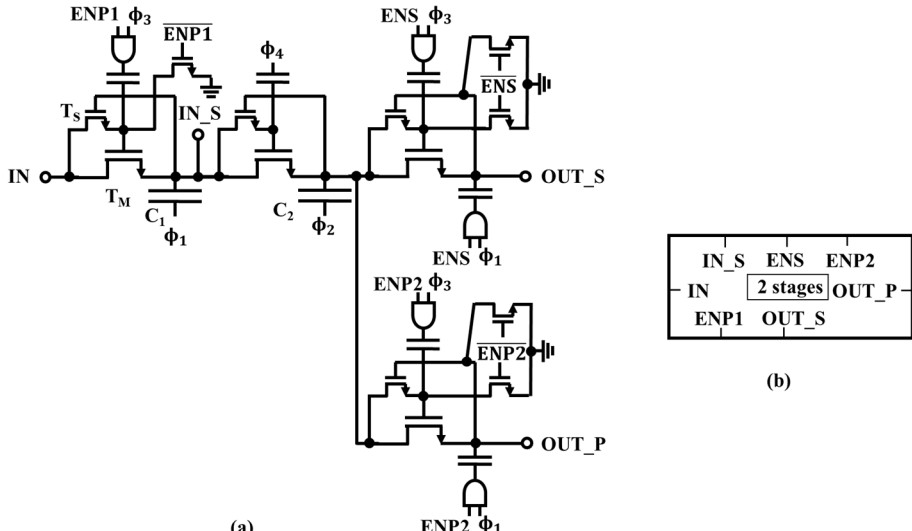

**Figure 4.** (**a**) Two-stage unit; (**b**) symbols of the two-stage unit.

Using this structure, one can configure two two-stage units (see Figure 5a) connected in series as shown in Figure 5b or in parallel as shown in Figure 5c [20,21]. When two two-stage units are connected in series, CP has a single array of four stages. When two two-stage units are connected in parallel, CP has two stages with twofold larger stage capacitance. As a result, the former configuration has higher maximum attainable output voltage and higher output resistance than the latter. Thus, the rise time of the output voltage can be reduced when the latter configuration is set to increase the output current while the output voltage is low, and the former configuration is set to increase the output voltage while the output voltage is high [20]. Another use case of this reconfiguration is that the load is varied such that a high output current at a low output voltage is required in the first operation and a low output current at a high output voltage is required in the second operation [21].

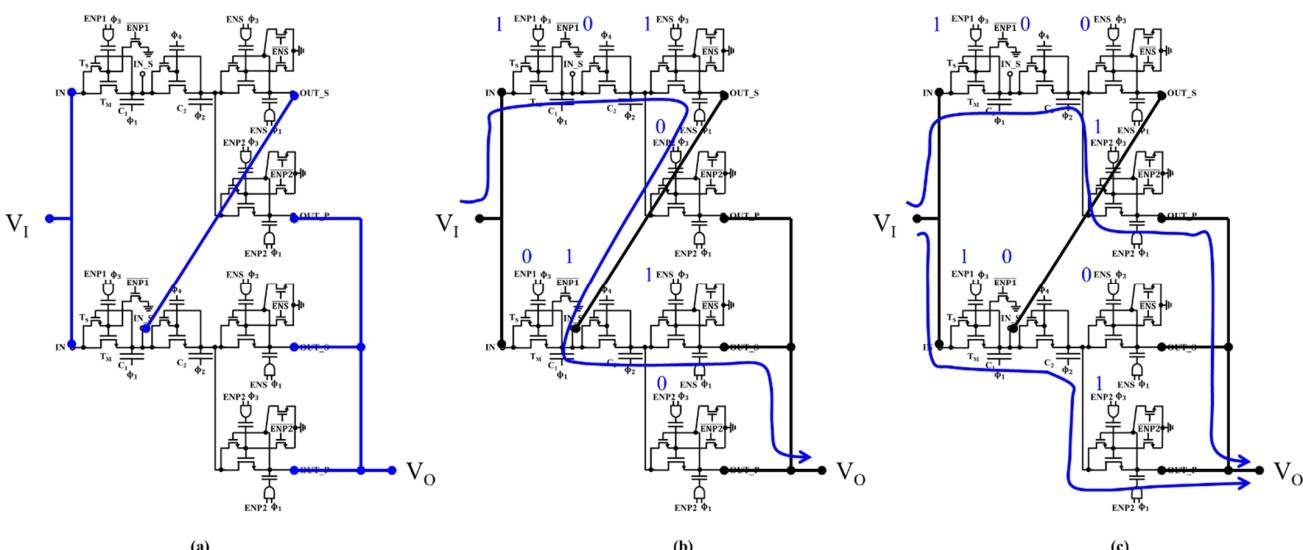

**Figure 5.** (**a**) Reconfigurable CP with two two-stage units; (**b**) operation of the CP with two two-stage units connected in series; (**c**) operation of the CP with two two-stage units connected in parallel.

## 3. Concept of 1D MPPT

In this paper, MPPT at the CP output is the focus. In [22], a circuit model for TEG-driven CP is developed to determine $C$ and $N$ to maximize the output current $I_{PP}$ at

$V_{PP}$ with a predetermined $f$. The model equation is expressed below, while the circuit parameters are defined in Table 1. A flexible type of TEG [23] is proposed in this work, which has a relatively high $R_S$.

$$I_{PP} = \frac{(N+1)(V_{OC} - V_{TH}) - V_{PP} - \delta V}{(N+1)^2 R_S + \frac{N}{fC} + \delta R}. \tag{1}$$

**Table 1.** Definition of circuit parameters.

|  | Parameter | Definition | Default Value |
|---|---|---|---|
| TEG | $V_{OC}$ | Open-circuit voltage as a function of temperature gradient | 0.5, 0.7, 1.0, 1.5, 2.0 V |
|  | $R_S$ | Output resistance | 600 Ω |
| CP | $C$ | Stage capacitance | TBD |
|  | $N$ | Number of stage capacitors | TBD |
|  | $f$ | Clock frequency | 10 MHz |
|  | $V_{PP}$ | Target output voltage | 3.0 V |
|  | $V_T$ | Effective thermal voltage of charge transfer switches (CTS) | 26 mV |
|  | $I_{SAT}$ | Saturation current of CTS | 40 nA |
|  | $\alpha_T$ | Ratio of top plate capacitance to C | 0.05 |
|  | $\alpha_B$ | Ratio of bottom plate capacitance to C | 0.1 |

An effective threshold voltage of charge transfer switches (CTS) $V_{TH}$, a loss in the voltage gain due to the parasitic capacitance $\delta V$, and an additional output resistance due to the parasitic capacitance $\delta R$ are given by Equations (2)–(4), respectively.

$$V_{TH} = V_T \ln\left( 4^{\frac{1}{N+1}} \frac{(1 + \alpha_T) f C V_T}{I_{SAT}} \right). \tag{2}$$

$$\delta V = (V_{PP} + (N+1)V_{TH})\{ N f C R_S (\alpha_T + \alpha_B + \alpha_T \alpha_B) + \alpha_T \} - \alpha_T V_{OC}. \tag{3}$$

$$\delta R = (\alpha_T + \alpha_B N^2) R_S. \tag{4}$$

Determination of the optimum combination of $C$ and $N$ for a given $V_{OC}$ was demonstrated in [22]. $f$ is set at the predetermined value which maximizes $I_{PP}$ at $V_{PP\_TGT}$ [24]. In other words, the predetermined value of $f$ can minimize the CP area to output a target $I_{PP}$ at $V_{PP\_TGT}$. To determine $f$, $I_{PP}$ at $V_{PP\_TGT}$ of 3.0 V was measured as a function of $f$ with SPICE. Four CP configurations, as discussed below in detail, were tested in the case of $(N, C)$ of (2, 160 pF), (4, 80 pF), (8, 40 pF), and (16, 20 pF), namely, 8-2, 4-4, 2-8, and 1-16 modes at $V_{OC}$ of 2.0 V, 1.5 V, 1.0 V, and 0.5 V, respectively, as shown in Figure 6a. Figure 6b shows $I_{PP}$ normalized by the maximum value in each configuration. Regardless of configuration, $f$ of 8–10 MHz gave the maximum output current. Thus, $f$ of 10 MHz was selected in this work.

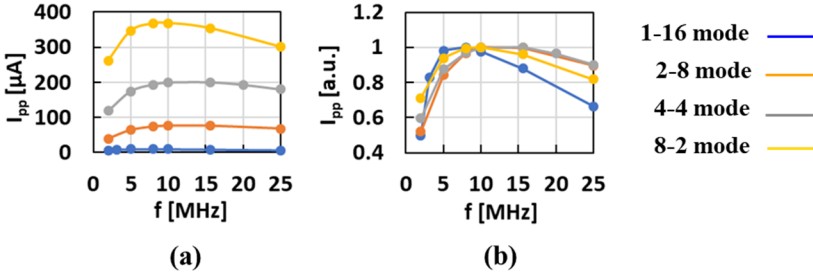

**Figure 6.** Clock frequency vs. output current at $V_{PP}$ of 3.0V. (**a**) absolute value; (**b**) arbitrary unit.

Unlike a given $V_{OC}$ in [22], how the optimum combinations of $C$ and $N$ vary with $V_{OC}$ was the concern in this paper. Figure 7 shows the contour plots of the output power $P_{OUT}$ over the $C$–$N$ plane in the case of $V_{OC} = 0.5$ V (a), 0.7 V (b), 1.0 V (c), 1.5 V (d), and

2.0 V (e) [25]. Points in red indicates the optimum combinations of $C$ and $N$, namely, $C_{OPT}$ and $N_{OPT}$, respectively, which enable CP to generate the largest $P_{OUT}$.

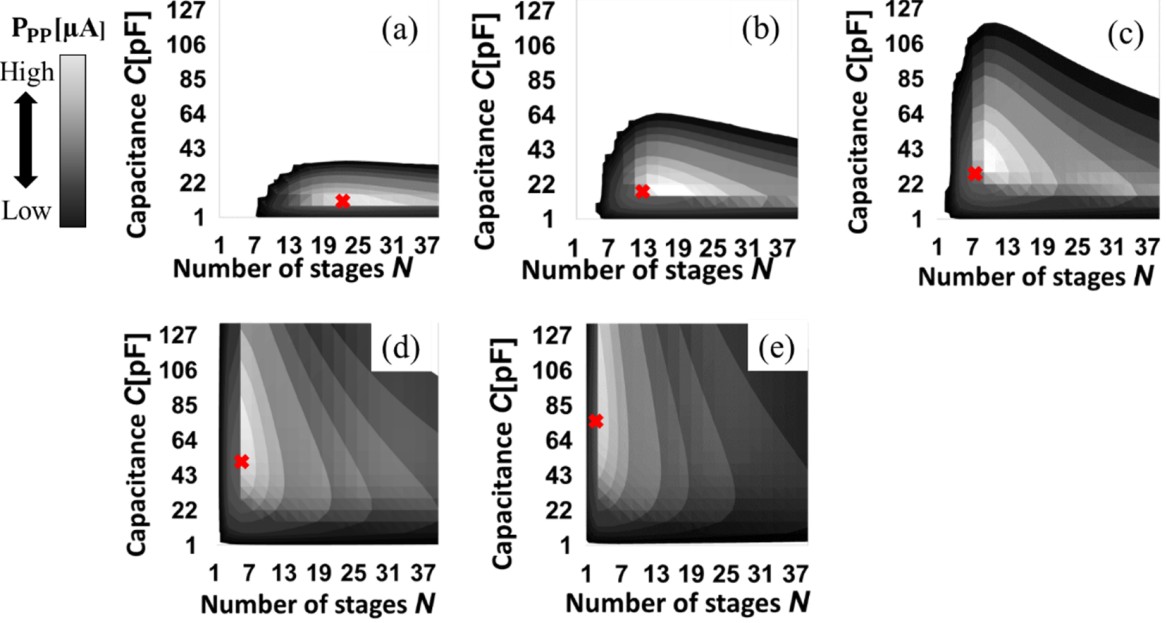

**Figure 7.** Contour plots of $P_{OUT}$ in case of $V_{OC}$ of 0.5 V (**a**), 0.7 V (**b**), 1.0 V (**c**), 1.5 V (**d**), and 2.0 V (**e**). "x" in each figure indicates the maximum power point.

Figure 8 shows how $C_{OPT}$ and $N_{OPT}$ vary with $V_{OC}$. The slope of the approximate line is $-1$, which suggests that their product, i.e., the CP area, should be constant. Intuitively, as $V_{OC}$ decreases, $N$ needs to increase to remain the voltage gain from the input to the output. If $C$ is unchanged, the input current should increase with larger $N$. This means that the input impedance would decrease. To keep the impedance matching at the interface between TEG and CP, $C$ needs to decrease as $N$ increases. Conversely, as $V_{OC}$ increases, $N$ must decrease whereas $C$ must increase to keep the voltage gain and the input impedance at the same time. To operate CP in MPPT at the CP output, one needs to design CP so that $N$ can vary while $C$ can vary inversely proportional to $N$ when $V_{OC}$ varies. As a result, the following functionalities are needed: (1) periodical detection of $V_{OC}$, (2) determination of $C$–$N$ combination for the present value of $V_{OC}$, and (3) reconfiguration of CP to have $C_{OPT}$ and $N_{OPT}$ for the present value of $V_{OC}$. In Section 4, the design is demonstrated. This procedure can be called a one-dimensional MPPT because one only needs to determine the combination of $C_{OPT}$ and $N_{OPT}$ for the present value of $V_{OC}$.

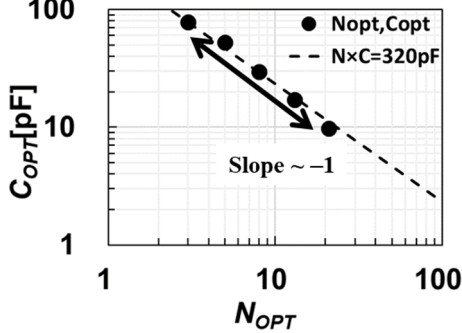

**Figure 8.** Relationship between $C_{OPT}$ and $N_{OPT}$ for TEG with $R_S = 600\ \Omega$ and $V_{OC} = 0.5$–2.0 V.

## 4. Circuit Design

### 4.1. Reconfigurable CP

In [15], a reconfigurable CP with fine-tuning capability to allow *N* of 1, 2, 3, 4, or 5 was proposed. As a result, 12 capacitors and 88 switches are needed for the five-stage CP. Many switches increase parasitic capacitance to the stage capacitors, which can affect voltage gain and power efficiency. Instead, another reconfiguration approach [21,22] was used to minimize the area overhead in this paper. Sixteen switches are added to the original 16-stage CP to allow CP to have two, four, eight, and 16 stages depending on the measured value of $V_{OC}$, as shown in Figure 8. Figure 9 shows a reconfigurable CP with eight two-stage units. With the signals in red are high and those in black are low, it can be reconfigured as a single-array 16-stage mode (1-16 mode), two-array eight-stage mode (2-8 mode), four array four-stage mode (4-4 mode), or eight-array two-stage mode (8-2 mode), as shown in Figure 9b–e. The lines in red show the conduction paths.

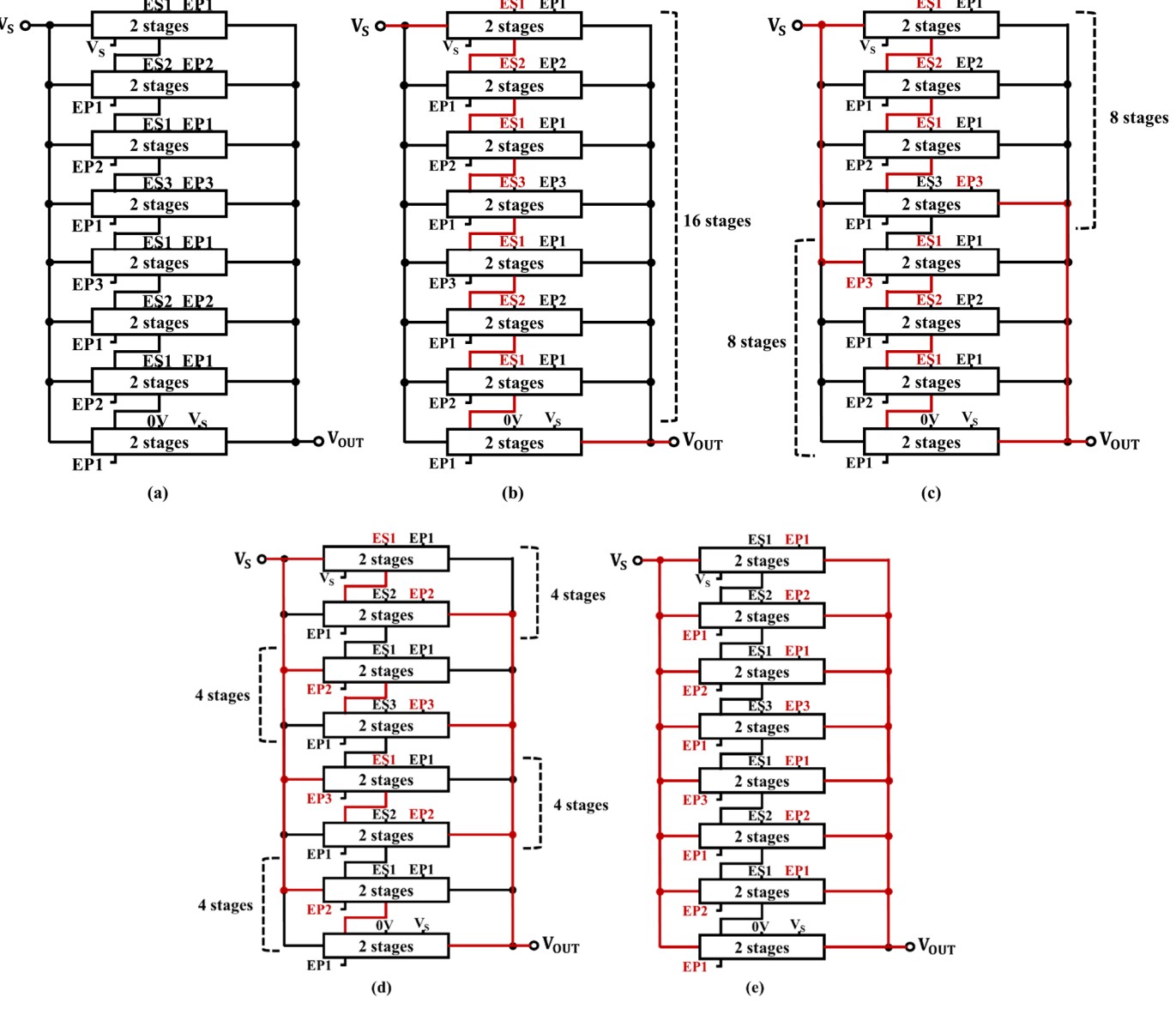

**Figure 9.** (**a**) Reconfigurable CP, (**b**) single-array 16-stage mode, (**c**) two-array eight-stage mode, (**d**) four array four-stage mode, (**e**) eight-array two-stage mode.

Figure 10a–e show $P_{OUT}$, $V_S$, $P_S$, $\eta_{CP}$, and $\eta_{CP\_MPPT}$ as a function of $V_{OC}$, respectively, in different modes. The aim of this work was to achieve MPPT at the CP output. According

to Figure 10a, the boundaries in $V_{OC}$ between 1-16 and 2-8 modes, between 2-8 and 4-4 modes, and between 4-4 and 8-2 modes are 0.55 V, 1.05 V, and 1.80 V, respectively. When the CP operates with 1-16, 2-8, 4-4, and 8-2 modes in $V_{OC} < 0.55$ V, 0.55 V $< V_{OC} < 1.05$ V, 1.05 V $< V_{OC} < 1.80$ V, and 1.80 V $< V_{OC}$, respectively, one can maximize $P_{OUT}$ regardless of $V_{OC}$. Figure 10b shows $V_S$ under the CP operation in MPPT. As suggested in [22], $V_S$ in cases where CP operates in MPPT for the output ($V_{MPPT\_OUT}$) is basically larger than that in cases where CP operates in MPPT for the input ($V_{MPPT\_IN}$), even though there are tiny ranges in $V_{OC}$ where $V_{MPPT\_OUT} < V_{MPPT\_IN}$. Figure 10c indicates that the input power to the CP converses as $V_{OC}$ increases. As a result, power efficiency $\eta_{CP}$ or $\eta_{CP\_MPPT}$ is maximized, as shown in Figure 10d or Figure 10e, where $\eta_{CP}$ and $\eta_{CP\_MPPT}$ are defined by Equations (5) and (6), respectively [15].

$$\eta_{CP} = \frac{P_{OUT}}{P_S}. \tag{5}$$

$$\eta_{CP\_MPPT} = \frac{P_{OUT}}{P_{AV}}. \tag{6}$$

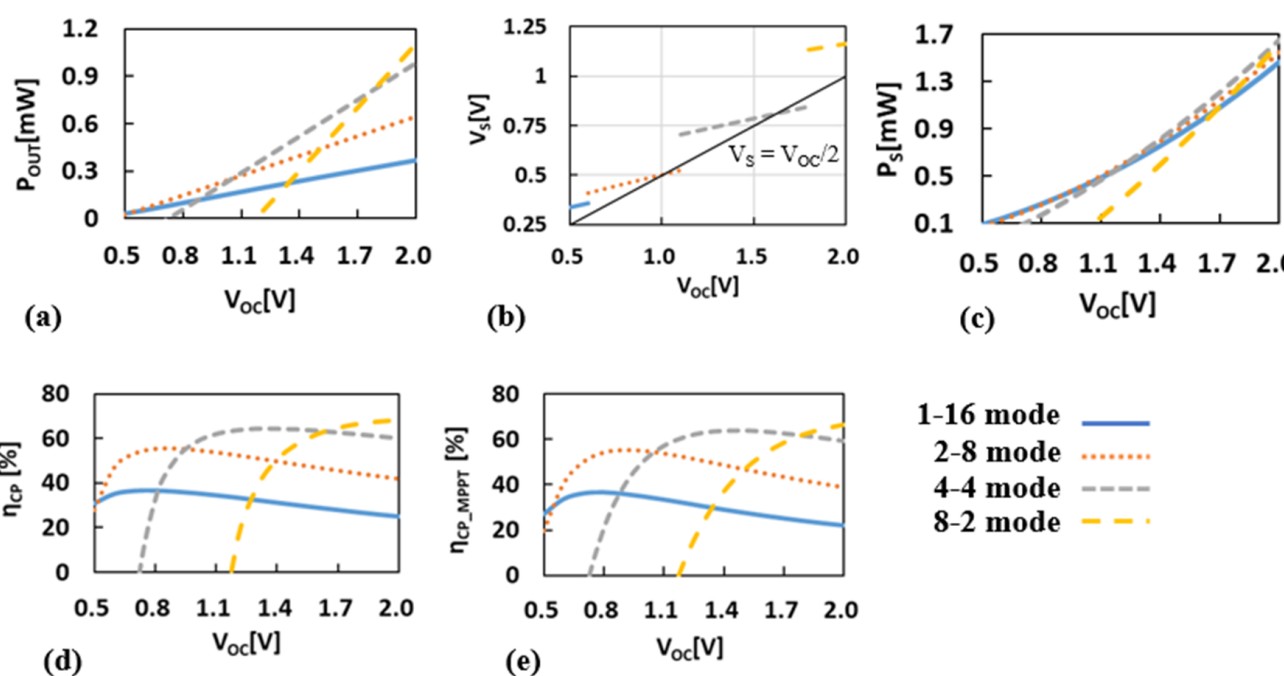

**Figure 10.** $P_{OUT}$ (**a**), $V_S$ (**b**), $P_S$ (**c**), $\eta_{CP}$ (**d**), and $\eta_{CP\_MPPT}$ (**e**) as a function of $V_{OC}$ in different modes.

$P_{AV}$ is the maximum attainable power of TEG when the impedance at the interface between TEG and CP is matched, as defined by Equation (7).

$$P_{AV} = \frac{V_{OC}^2}{4R_S}. \tag{7}$$

Figure 11a–d show $P_{OUT}$, $\eta_{CP}$, $\eta_{TEG}$, and $\eta_{CP\_MPPT}$ as a function of $V_{OC}$, respectively, in MPPT and fixed 1-16 modes. $\eta_{TEG}$ is defined by Equation (8) showing how much power is actually input to the CP normalized by $P_{AV}$.

$$\eta_{TEG} = \frac{P_S}{P_{AV}}. \tag{8}$$

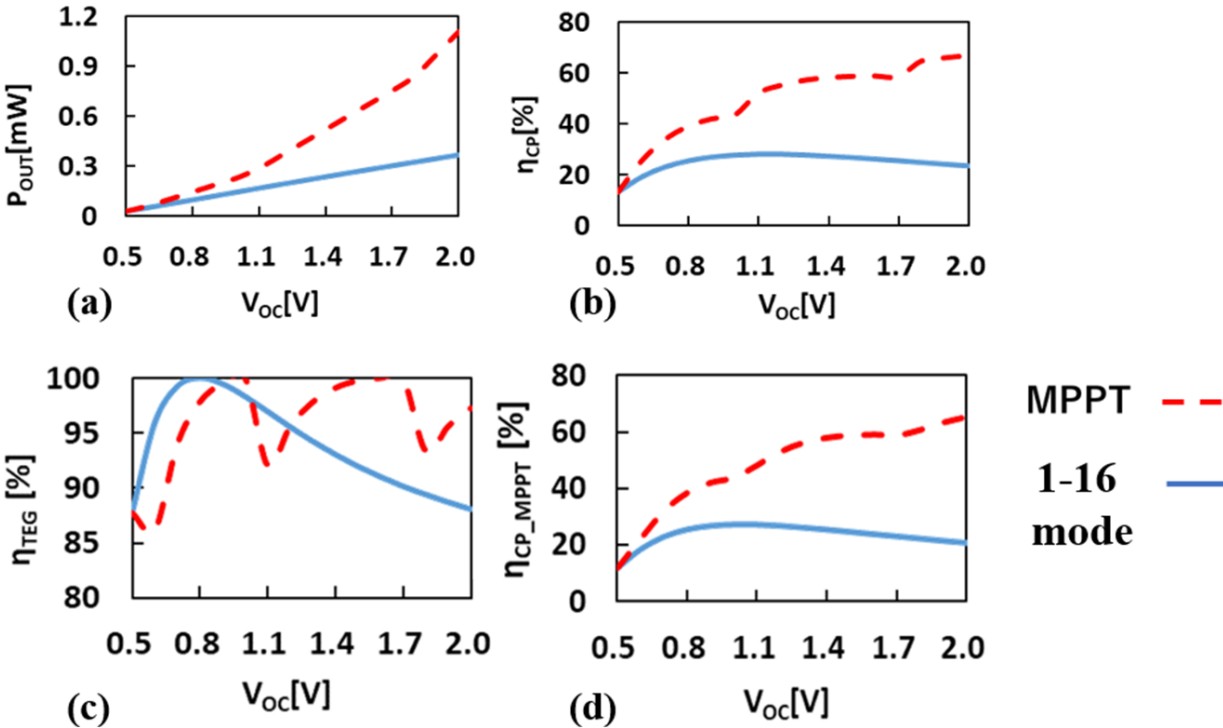

**Figure 11.** $P_{OUT}$ (**a**), $\eta_{CP}$ (**b**), $\eta_{TEG}$ (**c**), and $\eta_{CP\_MPPT}$ (**d**) as a function of $V_{OC}$ in MPPT and a fixed 1-16 modes.

The monotonic increase in $\eta_{CP\_MPPT}$ with MPPT indicates that this simple circuit structure with binary steps in $N$ can be sufficient with respect to system power efficiency. As a result, the average CP output power increases by a factor of 2.3 with the proposed MPPT when $V_{OC}$ varies in a rage of 0.5 V and 2.0 V randomly.

*4.2. System Design*

Figure 12a,b show the CP system and $V_{PP}$ waveform in ramping up, calibration, and user modes, respectively [26]. $V_{OC}$ can be measured when $V_{PP}$ stays high. Because the A/D converter (ADC) and bandgap reference (BGR) are powered by $V_{PP}$, one cannot know a value of $V_{OC}$ until $V_{PP}$ goes high. As a result, CP is set to the 1-16 mode initially, which can be boosted up to a target $V_{PP}$ of 3 V even with low $V_{OC}$. In the above demonstration, $N_0$ and $C_0$ are 16 and 20 pF, respectively. Once $V_{PP}$ reaches a target of 3 V, a calibration mode starts. The oscillator to drive CP is disabled to increase the input impedance of CP sufficiently high. After the input voltage $V_S$ is saturated to be close to $V_{OC}$, ADC measures the $V_{OC}$ value for C/N selector to determine the logic values for the CP control signals such as $ES1$ and $EP1$. CP is reconfigured to the optimum one for the current value of $V_{OC}$. Even without no CP operation in calibration mode, a voltage droop in $V_{PP}$ can be sufficiently small with low power ADC and BGR and large $C_{OUT}$. In the following user mode, CP operates in the current configuration. For a given application, the temperature gradient of TEG drifts in a specific time. The next calibration should start earlier than that specific time, but it is often not necessary.

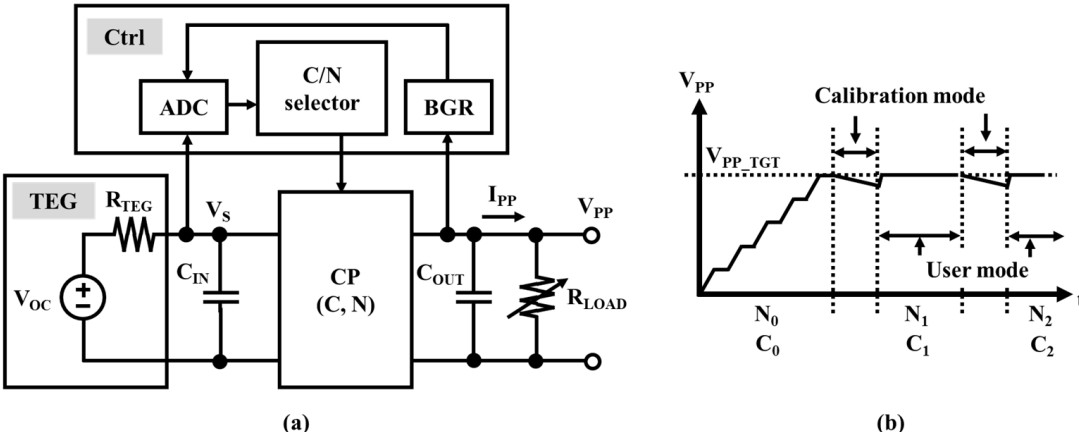

**(a)**　　　　　　　　　　　　　　　　　　　　**(b)**

**Figure 12.** CP system (**a**), $V_{PP}$ waveform in ramping-up, calibration, and user modes (**b**).

To validate the design, $V_{OC}$ shown in Figure 13a was input. $V_{OC}$ was varied from 0.5 V to 0.7 V and to 1.2 V. To save the simulation time, a step response in $V_{OC}$ was used. $V_{PP}$ was regulated at 3 V in 1-16 mode. After CP entered in steady state, a calibration signal was input with high in 100 ns. CP was reconfigured to 2-8 mode. Figure 13c shows 16 capacitor voltages in $T_{M1}$. Only eight signals are visible because the same stage voltages of two arrays of the eight-stage CP were overlaid. Similarly, Figure 13d shows 16 capacitor voltages in 4-4 modes. Only four signals are visible because the same stage voltages of four arrays of the four-stage CP were overlaid. The ripple in $V_{PP}$ increased from $T_{ST}$ to $T_{M2}$ with $I_{PP}$, which means that output power increased with $V_{OC}$.

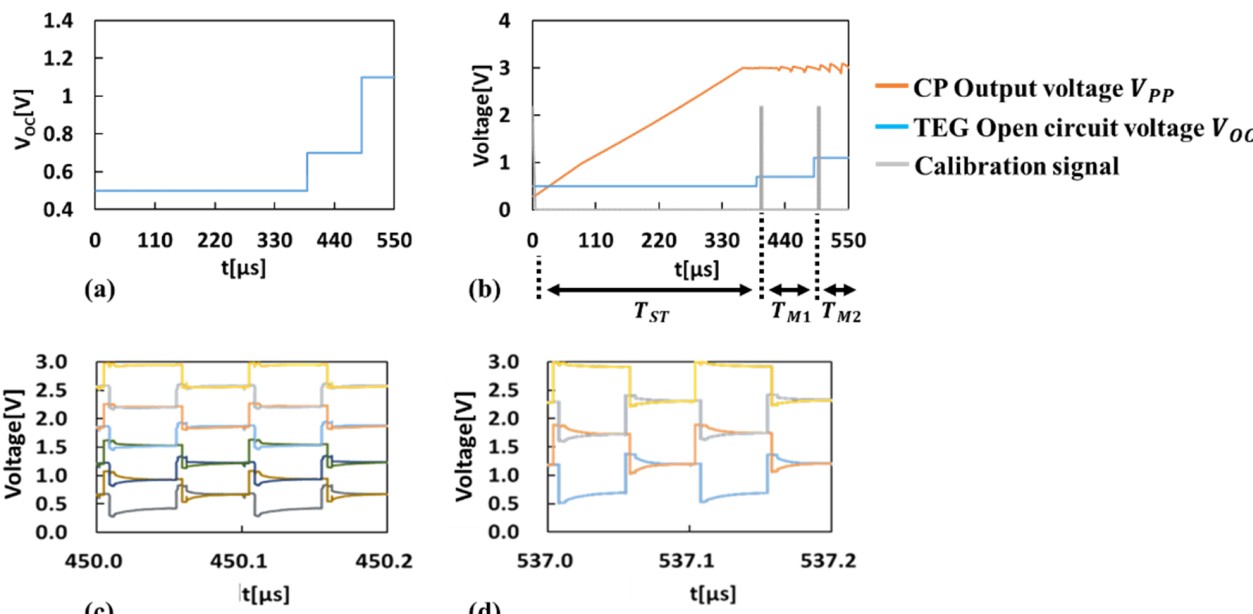

**Figure 13.** Input voltage $V_{OC}$ (**a**), $V_{PP}$, $V_{OC}$, and calibration signal (**b**), 16 capacitor voltages in $T_{M1}$ (**c**), and 16 capacitor voltages in $T_{M2}$ (**d**).

In the CP system shown in Figure 8, another feedback loop to disable CP when the input voltage $V_S$ lowers below a critical point where the output current becomes zero, as proposed in [11,12], was omitted for simplicity. Such a feedback loop is needed in a practical design.

### 5. Experiment

A part of the CP system was implemented in 65 nm CMOS to validate the design, as shown in Figure 14. Eight two-stage units were placed in order horizontally. CTS and small gate boosting capacitors were placed in the center. Clock and control signals were routed over the CTS region. Thus, additional circuit elements for CP to run with MPPT were minimal. The on-chip oscillator generated four clocks at 10 MHz. In this design, two-bit signals were input to select one among four configuration modes externally, instead of using ADC.

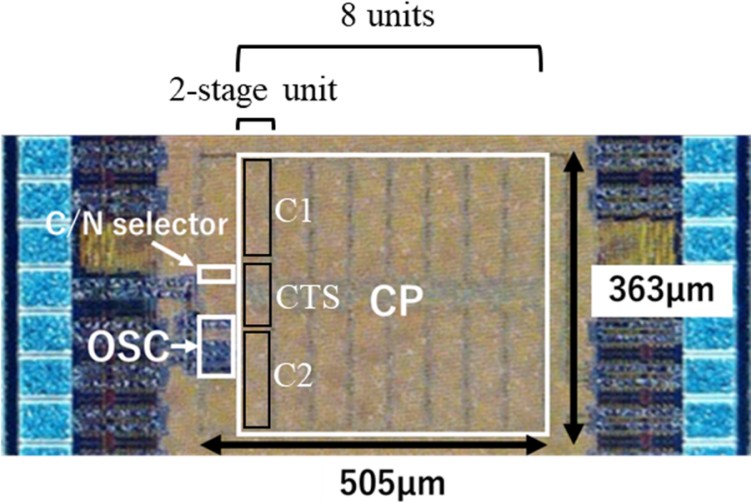

**Figure 14.** Die photo.

Figure 15 shows the measured $V_{OUT}$–$I_{OUT}$ at $V_S$ of 0.6V. $C_{IN}$ and $C_{OUT}$ of 1 nF were connected to the circuit. The slope in each mode was proportional to $fC/N$. The expected ratios between 1-16 and 2-8 modes, between 2-8 and 4-4 modes, and between 4-4 and 8-2 modes were as large as a theoretical value of 4 (a factor of 2 from $C$ and another factor of 2 from $N$). According to the measured maximum attainable output voltage, the effective threshold voltage was estimated to be 50 mV. Thus, $V_{OUT}$–$I_{OUT}$ curves in different modes were verified. Unfortunately, further measurement was not possible because all three fabricated dies were broken by accident.

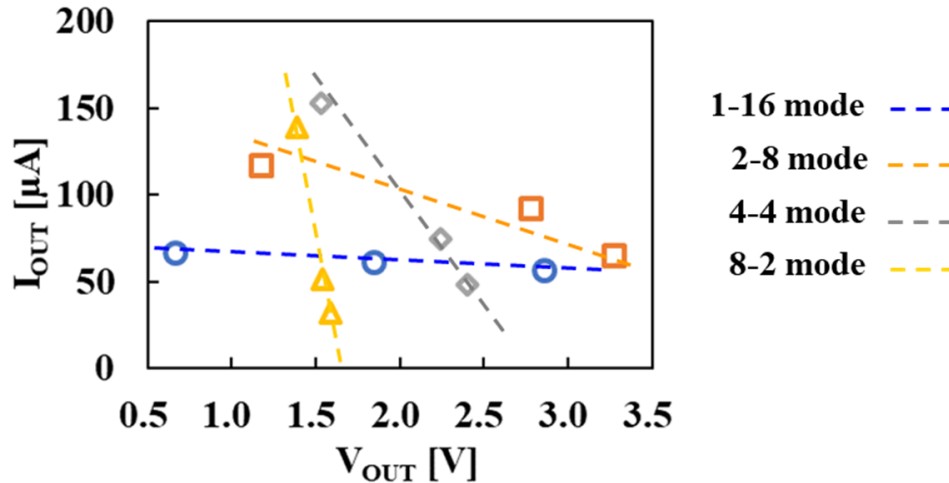

**Figure 15.** Measured $V_{OUT}$–$I_{OUT}$ at $V_S$ of 0.6 V in four CP modes.

## 6. Comparison with Previous Work

Table 2 compares this study with previous work [13–15]. In these previous studies, $f$ was used as a control circuit parameter to adjust the CP operation to MPPT. When one has room to decrease $f$, such a design needs larger $C$ than that of the CP, which is designed with an optimum $f$ for the minimum circuit area. As a result, the CP needs to prepare more area than the minimum. In this work, $f$ was fixed at 10 MHz regardless of $V_{OC}$. Therefore, the ratio of the size of CTS to $C$, whose optimum value was a function of f [18], was designed to be a single value. Thus, the two-stage unit can be commonly used for any configuration mode. Additional switches to change the connection state from serial to parallel or vice versa can be simple and implemented in the CTS region with a small area overhead. As a result, the output power density could be reduced in this study.

**Table 2.** Comparison with previous work.

| | | Liu, 2015 [13] | Bautista, 2016 [14] | | Yoon, 2018 [15] | This Work |
|---|---|---|---|---|---|---|
| Technology [nm] | | 180 | 180 | | 130 | 65 |
| Energy transducer | | TEG/PV (*1) | TEG/PV/MFC (*1) | | TEG | TEG |
| Operation range of $V_S$ [V] | | 1.28~3.0 | 0.25~1.1 | | 0.27~1.0 | 0.34~1.2 ($V_{OC}$ = 0.5~2.0 V) |
| Target output voltage $V_{PP\_TGT}$ [V] | | 3.3 | 1.8 | | 1.0 | 3.0 |
| MPPT | Procedure | Two steps in order | Two steps in parallel | | Two steps in order | Single step |
| | Parameters to be measured | $\Delta V_{PP}$, i.e., $I_{PP}$ | $V_S\sim V_{MPPT}$ | $V_{PP}\sim V_{PP\_TGT}$ | (1) $V_S\sim V_{MPPT}$, (2) $V_{PP}\sim V_{PP\_TGT}$ | $V_{OC}$ |
| | Parameters to be updated | (1) $N$, (2) $f$ | $f$ | $N$ | (1) Combination of $C$, $N$, (2) $f$ | Combination of $C$, $N$ |
| | Parameter to be maximized | $P_{OUT}$ | $P_{IN}$ | | $P_{IN}$ | $P_{OUT}$ |
| Area [mm²] | | 1.03 | 2.82 | | 0.835 | 0.302 (*2) |
| Maximum power efficiency of CP $\eta_{CP\_MAX}$ [%] | | 79 (TEG) 89 (PV) | 57 | | 64 | 67 |
| Maximum output power $P_{OUT\_MAX}$ [mW] | | 0.04 | 1.62 | | 0.40 | 1.11 |
| $P_{OUT\_MAX}$/Area [mW/mm²] | | 0.04 | 0.57 | | 0.48 | 3.66 (*2) |

(*1) PV: photovoltaic, MFC: microbial fuel cell; (*2) ADC to measure $V_{OC}$ is not included.

## 7. Summary

One-dimensional maximum power tracking was proposed to achieve both increased extracted power to the load and squeezed circuit area at the same time. A key finding in this paper is that MPPT at the output of CP was realized with the $C$–$N$ product, i.e., CP area, constant even at different open-circuit voltages of TEG. In calibration mode, $V_{OC}$ was measured with ADC while CP was disabled to determine an optimum CP configuration at the current $V_{OC}$. In the following user mode, CP was run with the updated reconfigured mode. By repeating this procedure periodically, CP can always stay under the MPPT condition. In the future, it will be verified whether the proposed MPPT method is applicable to other DC energy transducers such as photovoltaic and microbial fuel cells.

**Author Contributions:** Conceptualization, T.T.; methodology, K.N. and T.T.; software, K.N.; validation, K.N. and T.T.; formal analysis, K.N. and T.T.; investigation, K.N. and T.T.; writing—original draft preparation, K.N.; writing—review and editing, T.T.; funding acquisition, T.T. All authors have read and agreed to the published version of the manuscript.

**Funding:** This research was partially funded by Zeon Corp.

**Institutional Review Board Statement:** Not applicable.

**Informed Consent Statement:** Not applicable.

**Data Availability Statement:** Not applicable.

**Conflicts of Interest:** The authors declare no conflict of interest.

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
