# Peer review of "One-Dimensional Maximum Power Point Tracking Design of Switched-Capacitor Charge Pumps for Thermoelectric Energy Harvesting"

_electronics, doi:10.3390/electronics12051203_

Round 1

Reviewer 1 Report

This paper proposes one-dimensional (1-D) maximum power point tracking (MPPT) design which only requires measurement of one parameter (the input voltage of a switched-capacitor charge pump) for calibrating a power converter including the charge pump and thermoelectric generator. This paper is well written and interpreted. A minor revision is needed before acceptance for publication in this journal.

1.   Full name of “AC”, “IC” for the first mention should be indicated in the introduction part.

2.   In page 4 line 127, replace Reference [5] by the first author’s name and et al.

3.   In page 5 line 156, replace Reference [15] by the first author’s name and et al.

4.   References for Eqs. (5-8) should be added in the paper.

5.   Fig. 9 should be enlarged for clearer.

Reviewer 2 Report

This paper proposes one-dimensional maximum power point tracking (MPPT) design for calibrating a power converter such as the charge pump and thermoelectric generator. I have some comments that are expected to be addressed in the manuscript as follows

1. In the introduction section, the overview of the recent works should be expressed more detail with adding more references. Some sentences should be checked and clarified in the contribution of the paper.

2. System analysis should be provided.

3. The comparison with the existed method should be improved. In Table 2, it will be better with the explanation of parameters.

4. More simulation and experimental verifications can be provided.

Reviewer 3 Report

Authors present one dimension MPPT with increased power in reduced die area through exploiting C -N product rule. The topic is interesting and should be interested to readers. Few comments have been suggested  to improve the quality of the final manuscript as follows.

1. To include details control circuits for the reconfigural circuit in Figure 5. This could help readers easily to reproduce the circuit easily.

2. Insufficient of literature review. Critical review on previous research work about reconfiguralable CP etc should be included and summarised in a table.

3. Suggest to include more recent references.

Reviewer 4 Report

Koichi Nono and Toru Tanzawa report the theoretical and experimental demonstration of a novel design of MPPT where  power converter calibration occurs starting from single shot measurement.

Benefits of the reported design include minimal charge pump circuit area with simultaneous maximum  transferring current, as well as maximums charge pump output power during deltaT change.

While the performance of the proposed design is tested against a thermoelectric generator, potential applications to different DC energy transducer can be envisioned, including photovoltaics as well as microbial fuel cells.

The work is of interest, clear and well written and may represent a solid starting point for future developments. Elements of novelty emerge in the manuscript. The experimental demonstration reported in figures 10-11 is quite convincing.

Overall this is a solid piece of work which deserve publication in Electronics in its present form.

Author Response

Thank you for reviewing the manuscript.

Round 2

Reviewer 2 Report

Thank you for your responses.